# Land Use Changes in a Peri-Urban Area and Consequences on the Urban Heat Island

**Marianna Nardino** [1],*  **and Nicola Laruccia** [2]

[1]   National Research Council-Institute for Bioeconomy (CNR-IBE), Via Gobetti 101, 40129 Bologna, Italy
[2]   Emilia Romagna Region-Servizio Programmazione e Sviluppo locale Integrato, Viale della Fiera 8, 40127 Bologna, Italy; nicola.laruccia@regione.emilia-romagna.it
*   Correspondence: Marianna.Nardino@ibe.cnr.it; Tel.: +39-051-639-9001

**Abstract:** The effect of urbanization on microclimatic conditions is known as "urban heat islands". In comparison with surrounding rural areas, urban climate is characterized by higher mean temperature, especially during heat waves and during nights. This results in a higher energy requirement for air conditioning in buildings and in a greater bioclimatic discomfort for urban populations. The reasons of this phenomena are ascribable principally to the increase of solar radiation storage and to the decrease of dissipation of water by evapotranspiration in urban environment respect to rural ones. The aim of this paper is to give a quantification of the air temperature increase due to an urbanization process. This quantification is conducted by comparing surface energy balance (incoming and outcoming radiation and turbulent fluxes) in urbanized area versus rural areas. This quantitative approach will be validated using a fluidodynamic model (Envi-Met) in a case study area representative of one among the various regional models of urban area growth. In particular, the model of expansion of small towns around big cities (2003–2008 land use changes) of a plain near-urban area in the Po Valley region (Italy) was used.

**Keywords:** urban heat island; urbanization; urban surface energy balance; fluidodynamic modeling; Envi-Met

## 1. Introduction

The urbanization effect refers to a general increase in population and in the amount of industrialization of a settlement, due principally to the increase, in number and extent, of cities and the movement of people from rural to urban areas. The "urban sprawl" is used to define the increase in spatial scale or in the peripheral area of the cities. At the present, more than half of the global population lives in cities and cities themselves are growing to unprecedented sizes [1].

The high density of population and the consequent use of primary resources by urban residents, especially in the North Hemisphere, make cities and their inhabitants key drivers of global environmental changes [2,3].

Land-use and land-cover changes are recognized as causes of local, regional and global warming: The urban areas are the major sources of anthropogenic carbon dioxide emissions from the burning of fossil fuels for heating, from industrial processes, from transportation of people, etc. [4,5]. The main effects of land use changes (from rural to urban) can be found on the surface energy and water balances changes. The partitioning of sensible and latent heat fluxes is a function of varying soil water content and vegetation cover [6,7]. Water balance is strongly influenced by the soil sealing: Urbanization makes the surface permanently covered by impermeable artificial material (e.g. asphalt and concrete), for example through buildings and roads.

An increase of settlement areas over time is defined as land take, also referred to as land consumption. This process includes the development of scattered settlements in rural areas, the expansion of urban areas around an urban nucleus (including urban sprawl), and the conversion of land within an urban area (densification) [8,9]. Depending on local circumstances, a greater or smaller part of the land take will result in actual soil sealing. Urban sprawl can be defined as the unplanned incremental urban development, characterised by a low-density mix of land uses on the urban fringe. It is important to underline that even planned urban development may result in land take and soil sealing.

Soil sealing and realization of buildings on it alter the surface energy balance with two different main mechanisms:

(1) Soil sealing reduces the vegetation cover and prevents the storage of rainwater and consequently the amount of water stored into the soil. Consequently, water available for evapotranspiration processes is much lower than in a natural surface. It follows that the latent heat dissipated from urban surfaces is close to zero and the amount of advanced energy will be available for other processes usually related to an increase of the thermal field [10].

(2) Materials used for the buildings may, according to their specific thermal and optical properties, store energy and radiation in form of heat when the radiation budget ($R_n$) is positive and release energy when $R_n$ is negative.

The combination of these two different mechanisms results in a different thermal trend that is observed in the built environment compared to the surrounding rural areas.

One of the most prominent feature of the urban climate is the urban heat island (UHI) effect, which is strongly tied to the geometry and dimensions of building, land use patterns, vegetation cover and the intensity of the anthropogenic heat release [11,12]. The UHI makes the city warmer than its rural surroundings and it is stronger at night than during the day. This effect also decreases with increasing wind speed and cloud cover and it's less pronounced in summer and winter [11].

On average, urban temperatures may be 1–3 °C warmer than surrounding urban environment [6].

The aim of this work is to quantify the change in temperature range that can be expected on the basis of different energy balances resulting from the transformation of rural areas in urban residential or productive areas. The changes in the soil use (from natural surface to no permeable surface) returns an environment where the storage of heat during the day by building materials is released during the night, increasing the urban heat island. This effect is the main subject of the present work.

## 2. Materials and Methods

### 2.1. Energy Balance Method

The Earth surface radiation balance is described by the following equation:

$$R_n = (Sw_{in} - Sw_{out}) + (Lw_{in} - Lw_{out}) \tag{1}$$

where $R_n$ is the net radiation, $Sw_{in}$ is the incoming shortwave (visible) radiation, $Sw_{out}$ is the outcoming shortwave radiation, $Lw_{in}$ is the incoming longwave (infrared) radiation and $Lw_{out}$ is the outcoming longwave radiation.

The term $(Sw_{in} - Sw_{out})$ is the net shortwave radiation and it can be also described as $(1 - \alpha)Sw_{in}$, where $\alpha$ is the surface albedo used to quantify the solar radiation reflected by a surface. Albedo is a characteristic of the specific surface and depends on the optical characteristics of the reflecting surface.

$Lw_{in}$ and $Lw_{out}$ depend on the atmosphere and surface temperature following the black body equation:

$$Lw = \varepsilon \, \sigma \, T^4 \tag{2}$$

where $\varepsilon$ is the body infrared emissivity, $\sigma$ is the Stefan–Boltzmann constant ($5.67 \times 10^{-8}$ W m$^{-2}$ K$^{-4}$) and T is the body temperature expressed in K.

The net radiation given by (1) is then utilized to the partitioning processes at the surface, following the surface energy balance equation [12]:

$$R_n = H + LE + G \tag{3}$$

where H is the sensible heat flux, LE is the latent heat flux and G is the ground heat flux.

Usually the greatest part of the net radiation is used for sensible and evapotranspiration processes while the G term represent only the 10% of the total available energy ($R_n$).

To estimate the lower dissipation of latent heat flux due to the transformation of cultivated land in urbanized area, it is necessary to quantify the evapotranspiration (ET) before and after the transformation. The ET of a cultivated area depends on climatic characteristics of the site and on eventual water contributions from irrigation, on vegetable cover and on soil type. As vegetable cover, for this study, it was assumed the cultivation of winter wheat, not irrigated and growing in optimal pedological and climatic conditions to ensure a good water supply during vegetation season (November to June). Wheat is one of the more widespread dry crops in the Emilia-Romagna Region plain.

From a sealed surface, the evaporation of rain water can be considered negligible. On the other hand, from a winter wheat crop, in the case of optimal soil and climatic conditions, about 700 $lm^{-2}$ of water per year changes from liquid into vapor phase.

A very simplified, but amply adopted method to calculate potential evapotranspiration (ETc) of a crop consists on multiplying reference evapotranspiration ($ET_0$) per crop cultural coefficient (Kc):

$$ETc = Kc \, ET_0 \tag{4}$$

In this study $ET_0$ was calculated according to Hargreaves method [13] using the data of a meteorological station relatively close to the case study area, San Pietro Capofiume (44.648993 °N, 11.650055 °E, 11 m a.s.l.). The values of the cultural coefficient Kc for wheat were adopted following the [14] suggestions and reasonable Kc's for spontaneous grasses, growing after crop harvesting, were chosen.

If a land use change occurs, such as a significant urbanization, the terms of this budget equation change significantly and other terms, depending on different processes, must be taken into account. The Equation (3) of the surface energy balance can therefore be rewritten in this way:

$$R_n = H + LE + G + Qs \tag{5}$$

where Qs is the storage heat flux, which is strongly dependent on the ratio between green and sealed areas and on the building geometrical, optical and thermal characteristics.

Oke [6] proposed these equations for Qs:

$$day \, Qs = (0.20\lambda v + 0.33\lambda p) \, R_n + 3\lambda v + 24\lambda p \tag{6}$$

$$night \, Qs = (0.54\lambda v + 0.90\lambda p) \, R_n \tag{7}$$

where $\lambda v$ is the green area fraction and $\lambda p$ is the building area fraction.

The reconstruction of the thermal change given by the lower amount of evapotranspiration of a sealed surface was performed following the Fick law [12]:

$$\Delta T = Q \, Dz \, k^{-1} \tag{8}$$

where

Q = Latent Heat Flux + Storage Heat Flux (W $m^{-2}$)
Dz = reference height (2 m)
k = air thermal conductivity (0.026 W $m^{-1}$ $K^{-1}$ )

*2.2. Fluidodynamic Simulation (Envi-Met): The Case Study*

ENVI-met [15] is a three-dimensional non-hydrostatic microclimate model designed to simulate the surface–plant–air interactions within daily cycle in urban environment with a typical resolution of 0.5 to 10 m in space and 10 s in time.

The model has been widely used in many previous studies to simulated flow around and between buildings, exchange processes of heat and vapor at the ground surface and at the walls, turbulence exchange of vegetation and vegetation parameters, bioclimatology, and particle dispersion [16]. The model was validated as reported in [17].

In order to run the model, the detailed data on soil characteristics, buildings, vegetation, and initial atmospheric conditions for the area of interest were inserted.

After this input phase, the model is ready to run and the desired variables have been selected and saved into the output files [16].

ENVI-met can be used for several studies to test various urban canyon aspects as well as ratios and orientation effects on outdoor thermal comfort, the role of vegetation in the mitigation of the urban heat island effect, and other factors.

## 3. Results and Discussion

Hourly measurements of the surface radiation balance components ($Sw_{in}$, $Sw_{out}$, $Lw_{in}$, $Lw_{out}$) are available in stations relatively close to the case study area. San Pietro Capofiume was used as the representative station for rural open space. In this site CNR-IBE performed radiation components measurements for the period 1 January 2002–31 December 2003.

Bologna urban ARPAe (HydroMeteorological Service of the Emilia-Romagna Regional Agency for Environmental Protection) station (44.500754 °N, 11.328789 °E, 78 m a.s.l.) has been considered as the representative station for sealed conditions (measurements period 1 January 2006–31 December 2009). Although this data represents only few years, furthermore non-coincident, from Table 1 it can be inferred that the values of the two stations are substantially super imposable with regard to the $Sw_{in}$ and the $Sw_{out}$ values, while the differences between the values of $Lw_{in}$ and $Lw_{out}$ are attributable to the different surface types (rural and urban), principal aim of this paper.

**Table 1.** Annual mean values of the surface radiation balance components for rural (San Pietro Capofiume) and urban (Bologna urban) sites.

| Station | $Sw_{in}$ (W m$^{-2}$) | $Sw_{out}$ (W m$^{-2}$) | $Lw_{in}$ (W m$^{-2}$) | $Lw_{out}$ (W m$^{-2}$) |
|---|---|---|---|---|
| Bologna urban | 148 | 26 | 325 | 396 |
| San Pietro Capofiume | 156 | 27 | 305 | 361 |

Throughout the hourly measurements, in Table 2 are reported the diurnal and nocturnal $R_n$ values obtained for the two stations and for each month of the year.

**Table 2.** Annual mean of net radiation values in nocturnal and diurnal hours for rural (San Pietro Capofiume) and urban (Bologna urban) sites.

| Month | Nocturnal $R_n$ (W m$^{-2}$) | | Diurnal $R_n$ (W m$^{-2}$) | |
|---|---|---|---|---|
| | San P. Capof. | Bologna Urban | San P. Capof. | Bologna Urban |
| January | −17 | −39 | 78 | 52 |
| February | −31 | −57 | 138 | 124 |
| March | −34 | −63 | 253 | 201 |
| April | −28 | −62 | 246 | 278 |
| May | −24 | −52 | 288 | 280 |
| June | −36 | −60 | 287 | 285 |
| July | −51 | −71 | 279 | 293 |
| August | −39 | −67 | 255 | 239 |
| September | −28 | −62 | 198 | 189 |
| October | −26 | −55 | 157 | 141 |
| November | −18 | −44 | 77 | 71 |
| December | −19 | −47 | 68 | 48 |

Table 3 shows the values of mean monthly latent heat flux which would be missed in case of transformation of a wheat field into a sealed surface.

**Table 3.** Monthly mean values of reference evapotranspiration (ET$_0$), crop coefficient (Kc), crop potential evapotranspiration (ETc) and corresponding latent heat flux.

| Month | ET$_0$ (mm) | Kc (Wheat and Spon. Grass) | ETc (mm) | Latent Heat Flux (W m$^{-2}$) |
|---|---|---|---|---|
| January | 19.8 | 0.4 | 7.9 | 6.9 |
| February | 36.3 | 0.5 | 18.2 | 17.4 |
| March | 66.9 | 0.7 | 46.8 | 40.8 |
| April | 109.1 | 0.9 | 98.2 | 88.4 |
| May | 153.1 | 1.1 | 168.4 | 146.7 |
| June | 178.4 | 1.0 | 178.4 | 160.7 |
| July | 204.0 | 0.2 | 40.8 | 35.6 |
| August | 166.1 | 0.3 | 49.8 | 43.4 |
| September | 110.8 | 0.5 | 55.4 | 49.9 |
| October | 64.3 | 0.4 | 25.7 | 22.4 |
| November | 31.0 | 0.4 | 12.4 | 11.2 |
| December | 19.1 | 0.4 | 7.7 | 6.7 |

The area chosen as a case study is located in San Giovanni Persiceto (44.6283 °N, 11.1992 °E, 21 m a.s.l.), a small town close to Bologna city. The urbanization occurred in this area from 2003 to 2008 was estimated comparing aerial images (orthophoto by AGEA for 2008 and QuickBird satellite orthoimages by Digital Globe™–Telespazio for 2003) (Figure 1). The built-up surface varied from 5.5% in the 2003 to 30% in 2008 and the natural cover varied from 94.5% in 2003 to 70% in 2008. Inserting these values as λv and λp in (6) and (7), the difference of the storage heat flux Qs between 2003 and 2008 was obtained, separately for night and daytime hours. The annual difference in the cumulated storage heat flux is quite similar between night and day (+80 W m$^{-2}$ for the diurnal hours and −84 W m$^{-2}$ for nocturnal ones) which means that all the heat stored during the day by urban materials is re-emitted into the atmosphere during the night. On a monthly basis, this effect is visible in the winter months and during months with high energy availability (May, June, and July) the day energy accumulation is greater than the night emission (Table 4).

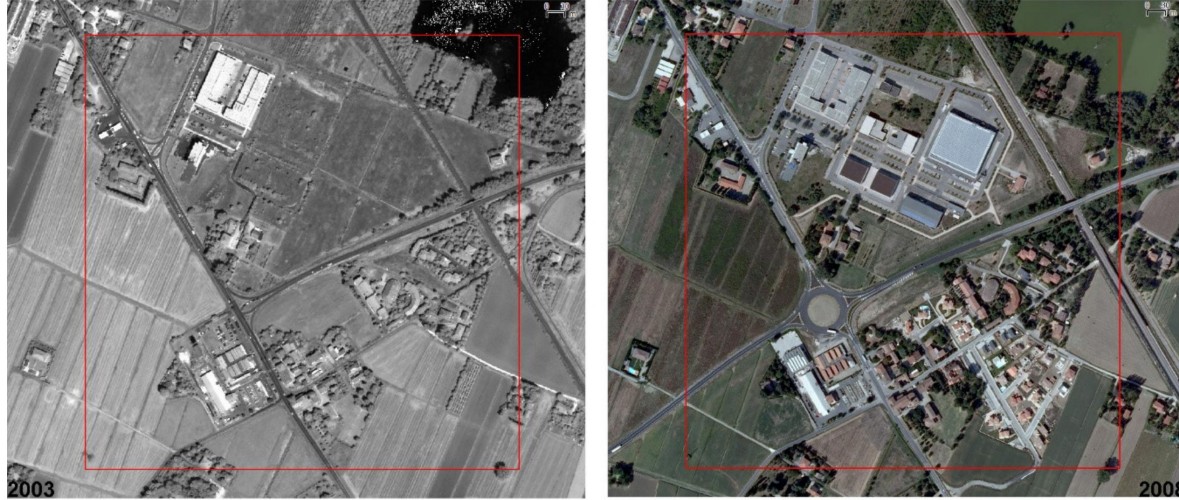

**Figure 1.** QuickBird satellite orthoimages (®Digital Globe™–Telespazio, 2003) (left) and aerial orthophotographs (AGEA) for 2008 (right).

**Table 4.** Monthly mean values of storage heat flux during diurnal and nocturnal hours for 2003 and 2008 and relative differences for each year.

| Month | Diurnal Qs (W m$^{-2}$) | | | Nocturnal Qs (W m$^{-2}$) | | |
|---|---|---|---|---|---|---|
| | **2003** | **2008** | **Difference** | **2003** | **2008** | **Difference** |
| January | 23.4 | 23.1 | −0.3 | −10.1 | −15.2 | −5.1 |
| February | 37.9 | 42.4 | 4.4 | −18.1 | −25.1 | −7.0 |
| March | 66.0 | 62.9 | −3.2 | −20.0 | −27.8 | −7.8 |
| April | 64.5 | 83.6 | 19.2 | −16.6 | −24.6 | −8.0 |
| May | 74.7 | 84.0 | 9.3 | −14.5 | −21.1 | −6.6 |
| June | 74.3 | 85.4 | 11.1 | −20.6 | −27.8 | −7.1 |
| July | 72.6 | 87.5 | 14.9 | −29.4 | −37.1 | −7.7 |
| August | 66.7 | 73.0 | 6.3 | −22.5 | −30.6 | −8.1 |
| September | 52.7 | 59.9 | 7.2 | −16.6 | −24.6 | −8.0 |
| October | 42.5 | 46.9 | 4.4 | −15.3 | −22.3 | −7.0 |
| November | 22.9 | 28.3 | 5.4 | −10.9 | −16.6 | −5.8 |
| December | 20.8 | 22.1 | 1.3 | −11.7 | −17.8 | −6.2 |

Since the evapotranspiration processes occur only during diurnal hours, the difference between the two reference years in terms of latent heat flux is assigned entirely to daylight hours. The greater energy availability for 2008 for heat transfer processes was obtained summing the diurnal latent heat flux differences (proportioned according to the fractions of cultivated and urban areas in 2003 and 2008) with the differences in diurnal storage heat flux. Nocturnal thermal variation depends only on variations in storage heat flux reported in Table 4.

Table 5 illustrates the total monthly differences (2008 vs 2003) in energy budget for diurnal hours and the corresponding diurnal, nocturnal and mean temperature variations calculated according to (8).

The largest increases in temperature due to an urbanization process occur during the summer months, when the energy available for exchange processes is greater.

This approach is an estimate of changes in energy balance and this leads at uncertainties, but in general the results show an annual mean increment in air temperature of 0.36 °C due to this urbanization process. Again, the single months show the differences due to energy availability, giving greater values of air temperature increment during summer months.

**Table 5.** Total difference (2008 vs 2003) in energy budget for diurnal hours and corresponding diurnal thermal variation (ΔT day), nocturnal thermal variation (ΔT night) and mean thermal variation (ΔT mean).

| Month | Δ LE + Diurnal Δ Qs (W m$^{-2}$) | ΔT day (°C) | ΔT night (°C) | ΔT mean (°C) |
|---|---|---|---|---|
| January | 2.1 | 0.11 | 0.27 | 0.19 |
| Febrary | 0.1 | 0.00 | 0.37 | 0.19 |
| March | 13.7 | 0.71 | 0.41 | 0.56 |
| April | 3.8 | 0.20 | 0.42 | 0.31 |
| May | 28.7 | 1.49 | 0.34 | 0.92 |
| June | 30.6 | 1.59 | 0.37 | 0.98 |
| July | −5.7 | −0.29 | 0.40 | 0.05 |
| August | 5.0 | 0.26 | 0.42 | 0.34 |
| September | 5.7 | 0.30 | 0.42 | 0.36 |
| October | 1.4 | 0.07 | 0.36 | 0.22 |
| November | −2.5 | −0.13 | 0.30 | 0.09 |
| December | 0.4 | 0.02 | 0.32 | 0.17 |

The same area was studied with the Envi-Met fluidodynamic model in order to have a modeling feedback of estimation method based on the surface energy balance.

The two input areas (2003 and 2008), shown in Figure 2, were inserted through the program ENVI-met Eddie, taking into account the geographical location, building dimensions, land use patterns. The vegetation was assumed to be winter wheat. The domain model consisted of a 140 × 140 × 20 grid with a spatial resolution of 5 m × 5 m × 2 m, resulting in a horizontal area of 700 × 700 m$^2$ with a 40 m vertical extent.

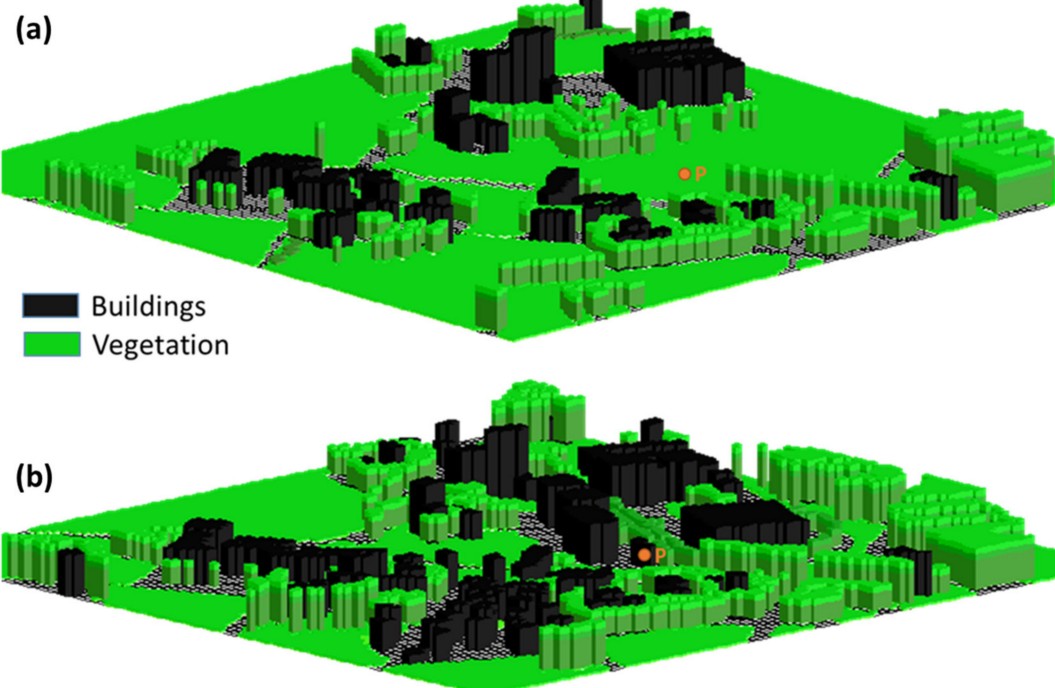

**Figure 2.** Input maps inserted into Envi-met model for 2003 (**a**) and 2008 (**b**).

The case study reported here focused on the simulation of a typical hot day whose values of meteorological variables were obtained considering the data of the nearest ARPAe weather station (San Pietro Capofiume).

The simulation started at 9:00 a.m. and lasted for 24 hours. The configuration file contained the atmospheric values:

- Speed and wind direction at 10 m: 1.4 m s$^{-1}$, 90 °;
- Surface roughness length (z0): 0.1 m;
- Air temperature: 301.7 K
- Specific humidity at 2500 m: 7 g water/kg air
- Relative humidity at 2 m: 50%

As first results the potential temperature (the temperature that a sample of air attains if reduced to a pressure of 1000 millibars without receiving or losing heat) to the environment difference between the two simulation years (2003 and 2008) were plotted (Figure 3) at 14:00 p.m. The urbanization process carried out from 2003 to 2008 led to a warming of some areas especially those in which it has had a greater overbuilding. In these areas the 2008 potential temperature at 1.6 m height is about 0.4–0.6 °C higher than 2003.

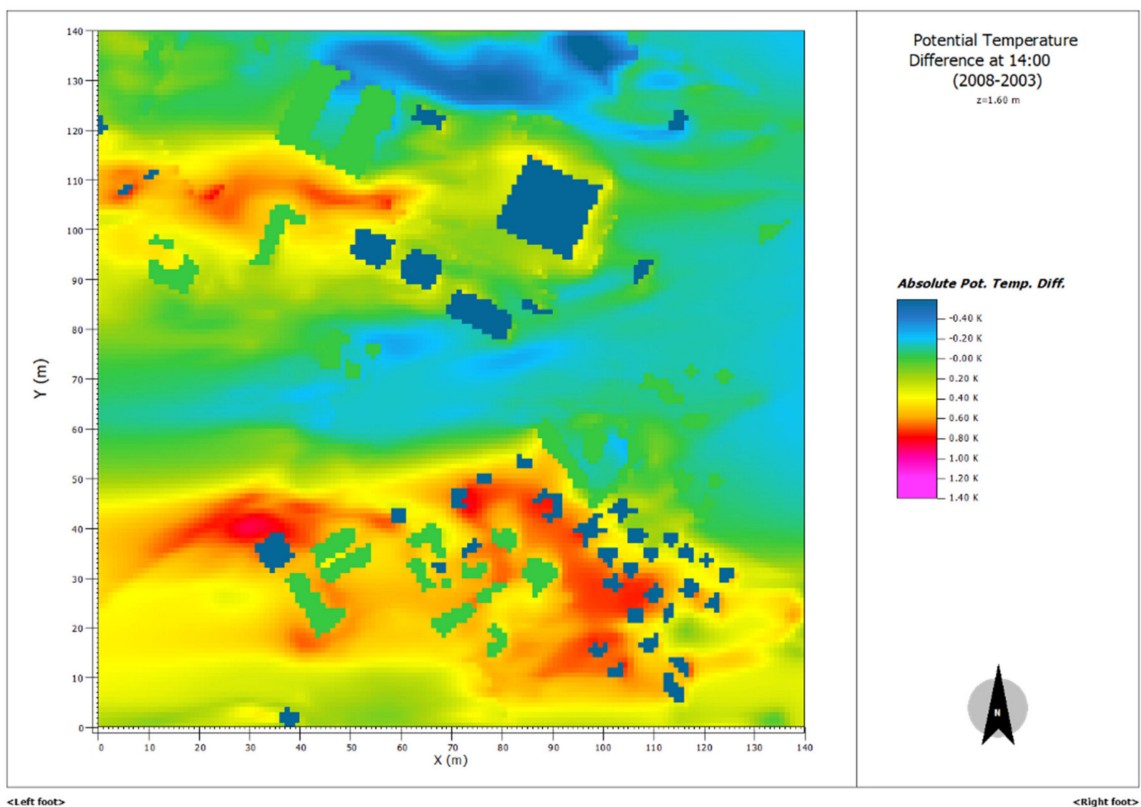

**Figure 3.** Potential temperature difference (2008–2003) at 14:00 pm at 1.60 m height for the considered case study.

During the night (Figure 4 at 2:00 a.m.) the potential temperature difference between the two years reaches 1 °C. This is typical of the urban heat islands, where the greatest effect is recorded during the night, so substantial increase occurs in daily minimum temperatures [18]. In addition, the total area tends to be warmer in 2008 than in 2003, and this effect is more evident during the night hours than daytime.

As a matter of fact, if the air temperature trends is plotted for a strategic point of the considered area (a point which was completely rural in 2003 and that became urban in 2008, marked as P in Figure 2) it can be seen that during the day air temperature tends to be the same for the two years and during the night the change turns out to be even 1 °C (Figure 5). The daily mean of air temperature differences is 0.35 °C that, looking at Table 5 for June month, is comparable with the night values, but not with the daytime ones. Probably the high available energy during this month means that further exchange processes come into play, that the balance method, used in this work, does not

account for. A better diurnal values estimate is surely necessary, even if during August and September (months similar to June as far as concerns air temperature values) the obtained delta temperature (0.34 °C and 0.36 °C) is very close to the model result.

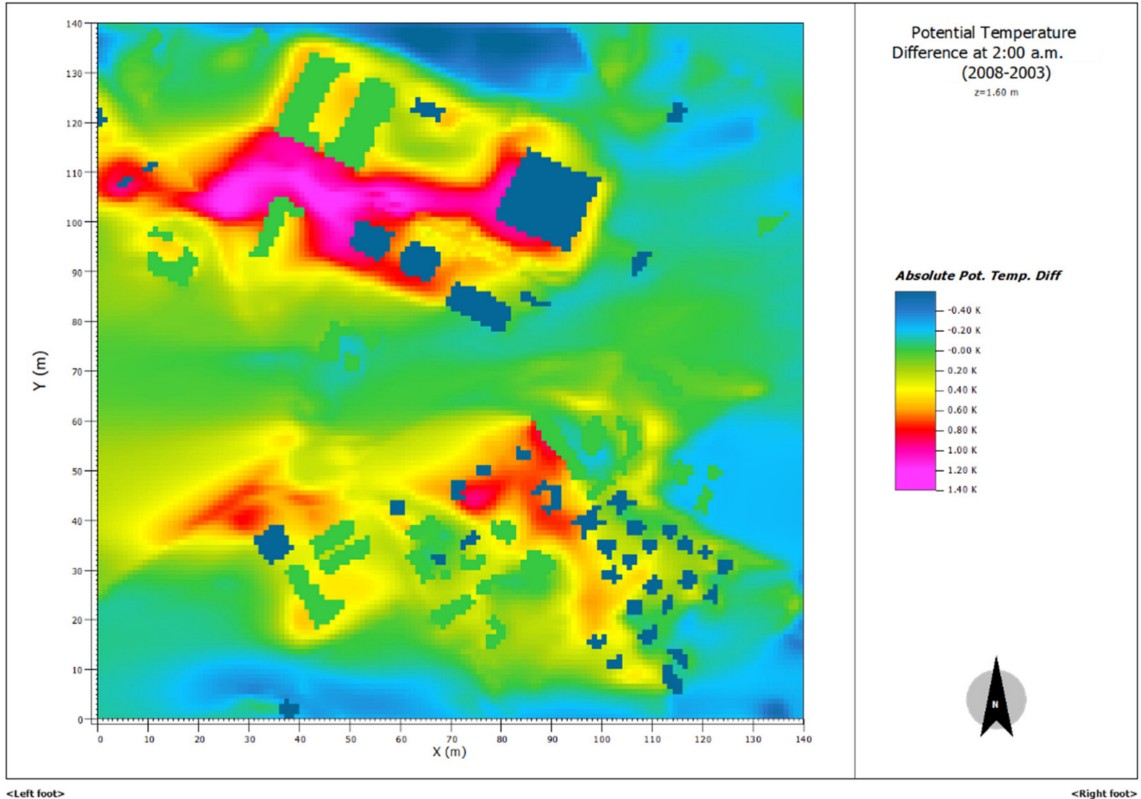

**Figure 4.** Potential temperature difference (2008–2003) at 2:00 a.m. at 1.60 m height for the considered case study.

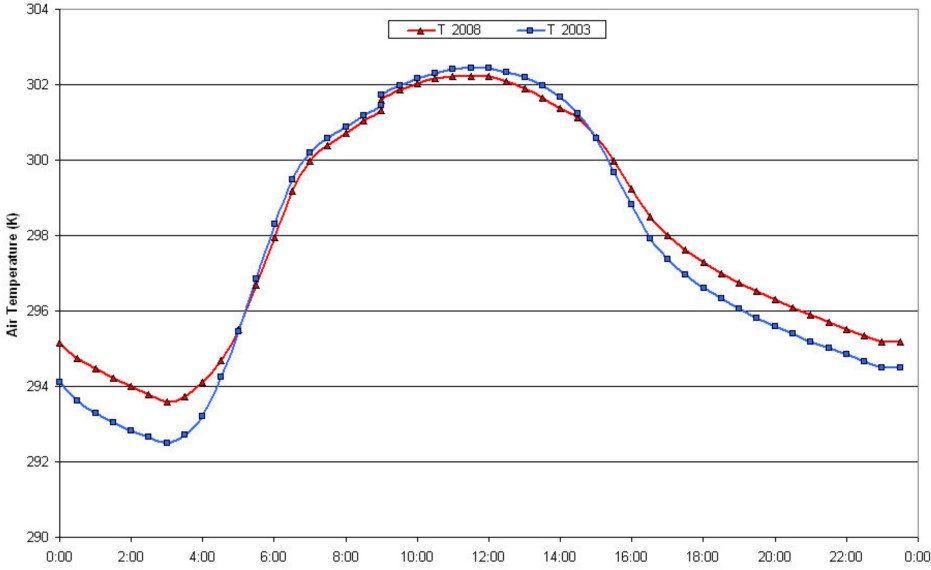

**Figure 5.** Air temperature for 2003 and 2008 in the P point marked in Figure 2.

June is the month in which the budget method assigned greater evapotranspiration values, but the fluidodynamic model does not consider specifically crop type and evapotranspiration needs. For this reason, the strongest disagreement results especially in this month.

On the other hand, the model results are in strong agreement with the annual average increase obtained with the method of the energy budget (0.36 °C).

This suggests a profitable use of this methodology to estimate the increase in air temperature due to urbanization processes on a regional scale.

## 4. Conclusions

The effect of an urbanization in a small town shows the consequences of the well-defined and studied "urban heat island": The impact of such kind of urbanization leads an increment in the air temperature, and therefore a strengthening of the urban heat island, close to 1 °C.

The energy balance method is verified and supported by a fluidodynamic model to have the right data interpretation. The idea is to develop a simple methodology to consider the urbanization effects in terms of temperature increase, economic cost that goes with it and biometeorological uncomfortable for the people.

This methodology could be applied at regional scale to improve and develop spatial planning but taking into considerations the mitigating effects strategies (i.e. urban green, special building materials, study of the orientation of buildings, and shadows).

The radiation and energy budget method utilized to calculate the air temperature differences due to an urbanization process showed some limits and some uncertainties.

Surely, the different approaches to compute evapotranspiration used in the budget method and in the fluidodynamic model and some hypothesis adopted (absence of water deficit in budget method), strongly influenced the results.

Anyway, the fluidodynamic model confirms an air temperature increment of the same magnitude order of the energy budget method, suggesting this last methodology can be considered a sufficiently reliable method to estimate and predict the variation in the thermal field due to changes in land-use. Its application to a regional scale should take into account the different climatic zones and the different models of urban growth (existing residential or production areas densification, expansion of urban area with new buildings, etc.). The storage heat flux computation can be improved considering further variables: volumetric ratio between sealed and unsealed surfaces and optical and thermal properties of building materials.

Land use change, and in particular urbanization process, results in air temperature increase. The methodology developed in this work can be improved to furnish grater information during a regional territory planning. Air temperature increment give a surplus of energy needs of air conditioning systems: throughout thermotechnical computations it is possible to obtain the economic cost to reinstate indoor air temperature after urbanization process.

Moreover, the bioclimatic discomfort for the population caused by urban land use increment can be computed and successively taken into account during the initial design of an urban area.

**Author Contributions:** The authors contribute at all in this study thanks to different thanks to the different skills: fluid dynamics and agronomics. The manuscript was written together.

**Funding:** This research received no external funding.

**Acknowledgments:** The authors would like to thank IdroMeteorological Service (ARPAe) of Emilia-Romagna Region for furnishing meteorological data through DEXTER service, AGEA for orthophotographs and Digital Globe™–Telespazio for QuickBird orthoimages, acquired through internal Map Service of Emilia-Romagna Region.

**Conflicts of Interest:** The authors declare no conflict of interest.

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
