# Peer review of "Land Use Changes in a Peri-Urban Area and Consequences on the Urban Heat Island"

_climate, doi:10.3390/cli7110133_

Round 1
Reviewer 1 Report
A review on the manuscript in journal Climate entitled “Land use changes in a peri-urban area and consequences on the urban heat island”.
This article attempts to analyze changes in air temperature caused by urbanization in peri-urban area based on changes in energy balance. Unfortunately, this task has not been the most successful. The article is not written in the best way.
Broad comments
A important disadvantage of the article is that the data and calculation methods underlying the study are assembled eclectically. For example, the components of surface radiation balance are compared for measurements taken in completely different years for rural and urban areas (for urban area measurements period 1/1/2006 - 31/12/2009; for rural area measurements 1/1 / 2002-31 / 12/2003). Solar radiation, and consequently radiation balance is highly variable in time and space, so comparing measurement data from different time at different locations is, to say the least, dubious.
Another disadvantage of the article is the fact that the formulas and explanations used in the formulation of the article use their own notation system, which makes it difficult to understand the content of the formulas and explanations. This is not in line with academic practice and refers to the fact that the authors of the article do not really have an understanding of the processes and regularities involved. Not using indexes (subscripts) makes the entire notation system incomprehensible. For example line 85 Rn = (Swin – Swout) + (Lwin – Lwout) may mean Rn = (Swin – Swout) + (Lwin – Lwout) or instead Rn = (Swin – Swout) + (Lwin – Lwout) or something else. Physical symbols in text and formulas should be corrected and entered using software tools (Equation editor, etc.).
Temperature and potential temperature are used in the article, but there is no explanation of how they are interrelated (no formula) and why these different parameters are used at all.
The text in lines 147-173 appears to be an extract/copy from the ENVI-met user manual, although no reference is made to it. What the user has to do is irrelevant. Matters is what the authors of the article did, what programs and data they used, what the data sources were, and so on.
Conclusions are short and vague, with no specific result or regularity. Although the title of the article includes the term "urban heat island", this is not mentioned in the conclusions.
Specific comments
Line 22 –“we used” is not an academic style. Excessive use of personal nouns [e.g., I, me, you, us] may lead the reader to believe the study was overly subjective. These words can be interpreted as being used only to avoid presenting empirical evidence about the research problem. Limit the use of personal nouns to descriptions of things you actually did.
The explanation of symbols used in the formula is not a new sentence and therefore begins with a lowercase letter. For example line 85 “Where Rn is the net radiation,…” must be “where Rn is the net radiation,..”. Line 127 same problem.
Lines 244-246 “The domain model consisted of a 140 x 140 x 20 grid with a spatial resolution of 5 m x 5 m x 2 m, resulting in a horizontal area of 140 x 140 m2 with a 40 m vertical extent.” cannot be right because 140 x 5 is not 140.
Lines 237-238 “Nocturnal thermal variation depends only on variations in storage heat flux reported in Table 4.” seems to be part of the text, not part of the table title.
Lines 247-248 - parts of the figures are denoted (fort example a), b) or otherwise) and referenced by symbols. The reference (top, bottom) would not be used for referencing.
Line 269 “…temperatures [15] In addition…” there is no dot at the end of the sentence.
Line 274 “…air temperature tends…” – “tends” or “trends”?
It must always be a space between the numerical value and unit symbol except the plane angle and percent. Lines 275, 276, 280, 281 - there is no space in front of the unit.
The capture of Figure 9 is basically wrong. The figure shows the change of air temperature rather than the change of temperature over a period of time (trend), such as a K/Year or K/hour.
Line 134 reference “(Stull 1988)” - referencing should use the same style throughout the article.
Line 129 - starting a sentence with a reference number is not a good style.
Reviewer 2 Report
The approach of this paper is well founded and the results are interesting. However, this manuscript lacks a discussion and needs substantial revision before it gets published.
Major comments:
Many statements in the introduction section needs references. For example: L37-39: Some suggested references- Lim et al., 2006; Li et al., 2015 L39-42: Some suggested references- Fischer et al., 2007; L45-48: Some suggested references- Liu, 2018; Liu et al., 2019
References:
Lim, Y. K., Cai, M., Kalnay, E., Zhou, L., 2006. Observational evidence of sensitivity of surface climate changes to land types and urbanization. Geophysical Research Letters. 32, L22712.
Li, Y., Zhao, M., Motesharrei, S., Mu, Q., Kalnay, E., Li, S., 2015. Local cooling and warming effects of forest based on satellite data. Nature communications. 6, 6603.
Fischer, E. M., Seneviratne, S. I., Vidale, P. L., Lüthi, D., Schär, C., 2007. Soil moisture-atmosphere interactions during the 2003 European summer heat wave. Journal of Climate. 20(20), 5081-5099.
Liu, Y., 2018. Introduction to land use and rural sustainability in China. Land Use Policy, 74, 1-4.
Liu, Z., Liu, Y., Baig, M. H. A., 2019. Biophysical effect of conversion from croplands to grasslands in water-limited temperate regions of China. Science of the Total Environment, 648, 315-324.
L72-74: The objective should be explicit. Rewrite into 2-3 sentences. L75-L81: Move this to discussion or conclusion section. L147-157: Don’t explain the model. Explain briefly what you have done or used. L158-166: May be deleted. L230: What is the range of the uncertainties. Explain in 2-3 sentences and explain how efficient your method over previous studies so that readers can get an overall reliability of this study with other studies. Some discussions or some insights are required. It could be just a short summary. For example, by outlining your scientific innovation in comparison to other papers. Figure 5: Are they really temperature trends? L297: Conclusions should be rewritten to describe objectives, findings and future scopes.
Minor comments:
L18-20: “will be” should be “is” L81: “~shadows,…” is not a proper way of writing. Replace it with “~shadows etc.” L85: The notations of the equation should be checked and corrected: “in” or “out” should be subscript. Same comment for all notations. L129: Rewrite the sentence and avoid to highlight the reference in the beginning. L140: “”Check the sentences properly. e.g., L163-L164: “…. however the dry bias is less than in the Grell AS.” Dry bias is not the Scheme, but in the simulated results. So it should be “…. however the dry bias is less than in the simulation with Grell AS.” L143-146: The sentence should be rewritten like “This model has been widely used in many previous studies to simulate flow around and between buildings,……………” L153, L261: Not clear. Mention the exact dates. L269: “~temperature [15]” should be “~temperature [15].” English in many sentences are not scientifically presented. Write your works in active or passive voice.
Round 2
Reviewer 1 Report
Broad comments
The authors of the article have made a number of improvements and additions that have improved the quality of the article.
However, there are a number of issues and problematic areas left in the article, the improvement of which makes it suitable for publication.
The most significant drawback is the treatment of air temperature by the authors of the article - it is completely eclectic. A systematic approach should be taken in this respect and the approach systematically organized. The temperature can be measured using the Celsius scale (symbol t, change Δt, unit °C), absolute temperature scale (symbol T, change ΔT, unit K), potential temperature (symbol θ, change Δθ, unit K). For example, line 137 uses t, unit K; line 222 used T, unit °C); line 241 air temperature unit K; line 278 unit °C; line 284 unit °C; etc. Line 247 "…potential temperature at 1.6 m height is about 0.4-0.6 °C higher…". The article does not discuss the calculation of the potential temperature and the purposes and needs for use.
Specific comments
Figure 1 and its capture require correction. The figure has two unmarked parts. Do you both have a picture of 2008?
Figure 5 capture needs to been changed. The figure shows the change of air temperature rather than the change of temperature over a period of time (trend), such as a K/Year or K/hour. “Diurnal air temperature cycles …” would be a more correct wording.
Sentence in lines 152-153 do not in any way match the content of the article by including an unrelated extract from the program manual.
The sentence in lines 150-151 should be reworded, you could use "were used" instead of "were inserted" (where ???). “In order to run the model…” is also a technical guide to using the program and is in no way related to the content of the study.
The article would look much better visually if all formulas were centered and their numbers aligned right.
Reviewer 2 Report
Authors have considered all the comments in the revised manuscript. It may be possible to accept the article with the following minor correction.
L38-19: References should appear in the order of increasing. It is not proper to cite [16,17] before [4]. Check all other places throughout the manuscript.
This is with my first reviewed comment. L237, L251, L269: What is 9 a.m. Is it universal time? Do authors mean everywhere in the world at 9 a.m. and throughout the year. Be specific. If this is your model start time or middle of the time, then avoid writing 9 a.m. Good choice could be at t=1, …. Or t=9 … Or something in this way.
Figure 8: What type of trend? Linear, exponential, Polynomial? Please mention. Otherwise “trend” word is misleading and can be kept as simple “Air temperatures”
